

# Dilatant normal faulting in jointed cohesive rocks: a physical model study

M. Kettermann[1], C. von Hagke[1], H. W. van Gent[1,a], C. Grützner[2,b], and J. L. Urai[1]

[1]Structural Geology, Tectonics and Geomechanics Energy and Mineral Resources Group, RWTH Aachen University, Lochnerstraße 4-20, 52056 Aachen, Germany
[2]Neotectonics and Natural Hazards, RWTH Aachen University, Lochnerstraße 4-20, 52056 Aachen, Germany
[a]now at: Shell Global Solutions International, Rijswijk, the Netherlands
[b]now at: COMET; Bullard Laboratories, Department of Earth Sciences, University of Cambridge, Cambridge, UK

Received: 10 December 2015 – Accepted: 11 December 2015 – Published: 14 January 2016

Correspondence to: M. Kettermann (michael.kettermann@emr.rwth-aachen.de)

Published by Copernicus Publications on behalf of the European Geosciences Union.

## SED

doi:10.5194/se-2015-131

## Dilatant normal faulting in jointed cohesive rocks

M. Kettermann et al.

Discussion Paper | Discussion Paper | Discussion Paper | Discussion Paper

## Abstract

Dilatant faults often form in rocks containing pre-existing joints, but the effects of joints on fault segment linkage and fracture connectivity is not well understood. We present an analogue modeling study using cohesive powder with pre-formed joint sets in the
upper layer, varying the angle between joints and a rigid basement fault. We analyze interpreted map-view photographs at maximum displacement for damage zone width, number of connected joints, number of secondary fractures, degree of segmentation and area fraction of massively dilatant fractures. Particle imaging velocimetry helps provide insights on deformation history of the experiments and illustrate the localization
pattern of fault segments. Results show that with increasing angle between joint-set and basement-fault strike the number of secondary fractures and the number of connected joints increases, while the area fraction of massively dilatant fractures shows only a minor increase. Models without pre-existing joints show far lower area fractions of massively dilatant fractures while forming distinctly more secondary fractures.

## 1   Introduction

Dilatant faults are ubiquitous features that occur at all scales in the upper crust. Most prominent large scale examples can be found at mid ocean ridges (Angelier et al., 1997; Friese, 2008; Sonnette et al., 2010; Wright, 1998), intra-plate volcanoes (Holland et al., 2006), continental rifts (Acocella et al., 2003), but also in cemented carbonates
and clastics (Ferrill and Morris, 2003; Moore and Schultz, 1999). They form major pathways for fluid flow, such as water, hydrocarbons, or magma, and consequently are of great interest for water and energy supply, geohazard assessment, and geodynamics (e.g. Belayneh et al., 2006; Caine et al., 1996; Crone and Haller, 1991; Ehrenberg and Nadeau, 2005; Gudmundsson et al., 2001; Lonergan et al., 2007). Several first order
models for the formation of dilatant fault networks exist (e.g. Abdelmalak et al., 2012; Abe et al., 2011; Acocella et al., 2003; Grant and Kattenhorn, 2004; Hardy, 2013; Hol-

**SED**

doi:10.5194/se-2015-131

**Dilatant normal faulting in jointed cohesive rocks**

M. Kettermann et al.

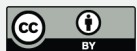

land et al., 2006; Holland et al., 2011; Kettermann and Urai, 2015; van Gent et al., 2010; Vitale and Isaia, 2014; Walter and Troll, 2001). However, the influence of pre-existing open joints on the formation of faults and fractures is largely untested, although this may have great influence on the fault's geometry and evolution (e.g. Butler, 1989; Gi-ambiagi et al., 2003; McGill and Stromquist, 1979; Schultz-Ela and Walsh, 2002; Virgo et al., 2014). This is also of interest for understanding fluid flow through the fault zone.

In this contribution, we focus on the influence of pre-existing joints on the formation of dilatant normal fault networks. In particular, we investigate the evolution of dilatant fault networks, which form at different angles with respect to a pre-existing layer-bound joint network. To this end, we performed a series of scaled analogue models. Our first goal is to quantify how the angle of pre-existing joints with respect to the active basement fault influences the opening behavior of the fault system. Quantifying this parameter will enable us to predict the evolution of segmentation, as well as orientation of secondary faults in the fracture network. In a second step we discuss our results in framework of natural examples. First, the fault network in the Canyonlands National Park, which showcases an open fracture network influenced by pre-existing joints (Fossen et al., 2010; Kettermann et al., 2015; Schultz-Ela and Walsh, 2002), and second, volcanic environments, in particular mid-ocean ridges as for example exposed in the rift zone in Iceland (Angelier et al., 1997), and caldera collapse in the Campi Flegrei, Italy (Vitale and Isaia, 2014).

## 2    Analogue modeling of dilatant faults in a jointed host rock

For our experiments we used the analogue device designed by Holland et al. (2011), which has a length, width, and depth of $28 \times 30 \times 19$ cm, respectively (Fig. 1). This box has a dip-slip half-graben geometry, with a basement fault dip of 60°, and maximum displacement is 4.5 cm. Throughout this article we quantify displacement as percent-age of sediment layer thickness. Therefore, the maximum displacement of 4.5 cm at a layer thickness of 19 cm translates to 23.7 % displacement. Modeling material as

**SED**

doi:10.5194/se-2015-131

**Dilatant normal faulting in jointed cohesive rocks**

M. Kettermann et al.

well as our experimental setup is based on previous analogue models of dilatant fault networks (Holland et al., 2006, 2011; van Gent et al., 2010). We used hemihydrate ($CaSO_4 \times 0.5\,H_2O$) powder because it has a well-known cohesion and tensile strength, and can develop vertical walls. Therefore, it is suitable to implement cohesionless joints into the models, and produce dilatant faults and open fractures. Properties of the material are well known (van Gent et al., 2010). The powder compacts easily, and increasing sieving-height leads to higher densities in the sandbox. This trend stops at a sieving-height of about 30 cm, at which the powder reaches a constant velocity due to a balance of air friction and gravity (Holland et al., 2011; van Gent et al., 2010). After sieving from a height > 30 cm, the powder has a density of $732\,kg\,m^{-3}$, and a porosity of 75 %. Tensile-strength is 9 Pa (method after Schweiger and Zimmermann, 1999) for the un-compacted powder, increasing proportionally to the pre-compaction stress. The cohesion derived from shear tests is about 40 Pa. Both tensile strength and cohesion increase with increasing compaction, i.e. overburden pressure or burial depth in the box. We scaled our experiments as explained by van Gent et al. (2010) and applying the laws derived by Schellart (2000). For example a model height of 19 cm represents approximately 600 m of sandstone in nature with a cohesion of 70 MPa.

As the powder is very sensitive to compaction, it is important to form joints without damage to the surrounding material. Initial test using a knife led to compaction of particles adjacent to produced joints (Fig. 2a and b). Minimum disturbances were achieved by mounting thin low friction paper sheets in the box with spacing of 2.5 cm prior to sieving. Removing the paper after filling the box leaves cohesionless open (< 1 mm aperture) joints without compacting or fracturing the surrounding material (Fig. 2c and d), and furthermore guarantees consistent depth of the joint. In order to reduce friction between the powder and the side-walls, paper sheets are mounted along the moving side-walls and removed before the beginning of the experiments. However, in some cases extraction of these paper sheets caused fractures orthogonal to joint strike at the outer edges of the experiment (i.e. close to the wall), visible before starting the experiment. These fractures may open during initial stages of the experiment, but do

not accommodate much strain, and do not influence fault geometry (see below). As these fractures are artifacts and can be followed throughout the experiments, we did not include them in the quantitative analyzes. The joints penetrate 5 cm deep into the powder (Fig. 1). We performed experiments with systematically increasing angles between the joints and the basement fault (0, 4, 8, 12, 16, 20, and 25°). The angle is in the following referred to as JF-angle.

In analogue models where no erosion is applied, deformation within the sandbox is reflected at the surface. A useful tool to measure surface evolution of analogue models is particle image velocimetry (PIV) (e.g. Adam et al., 2005). To enhance contrast, we added some sand grains to the hemihydrate powder at the top of the experiments. The small amount of sand ($\ll$ 1 vol. ‰) is assumed to have no influence on mechanics of the powder column or fault development. We recorded our experiments with two computer controlled DSLR cameras (Nikon D80 and D90 with resolutions of 10 and 12 million pixels, respectively), one in top view, and one in oblique view (Fig. 3). We use the top view photographs for PIV analysis, to identify areas of the model at which deformation localizes, and calculate the displacement fields. With this analysis, we detect which joints are reactivated at which state of deformation. The oblique view provides an optic impression of strain distribution on different joints and the 3D geometry of the model.

## 3   Analogue modeling results

We started our series with an experiment without pre-existing joints as a reference. In this experiment, the master fault shows a concave shape towards the hanging wall over the width of the box. This is a reasonably expected result as the fault that develops in our cohesive material is sub-vertical close to the surface, and thus substantially steeper than the predefined 60° fault dip of the sandbox. Close to the sidewalls of the box friction forces the powder to follow the 60° dip of the basement fault further towards the footwall. Where uninfluenced by sidewall effects, the fault forms as dilatant fault with vertical fault scarp close to the model's surface. The fault surface is rugged and

Discussion Paper | Discussion Paper | Discussion Paper | Discussion Paper |

**SED**

doi:10.5194/se-2015-131

**Dilatant normal faulting in jointed cohesive rocks**

M. Kettermann et al.

a small volume of rubble fills the opening gap at the fault (Holland et al., 2006; van Gent et al., 2010). A dense and interconnected network of secondary fractures parallel to the master fault forms gradually during fault evolution as a result of fault migration without a clear pattern but rather undulating in map-view. An antithetic fault forms as
well and shows the same type of migration and fracture network as the master fault. Overall we note that the observed fault and fracture pattern in homogeneous material is very different as compared to inhomogeneous experiments with pre-existing joints, as expected (Fig. 4). In the following we describe observations of the structural evolution of experiments with pre-existing joints including quantitative analyses of key parameters.
Figure 5 shows top view images and the corresponding PIV results for all experiments, which we will describe in the following. In order to identify and distinguish parts of the model that experience different amounts of deformation we show the total displacement vectors summing up the entire deformation until maximum displacement.

Our observations can be subdivided into two categories. First, features which can be
observed in all experiments, and develop after a similar amount of strain applied. Secondly, as opposed to these consistent features, we observe features that are variable, i.e. change with increasing angle between basement fault strike and joint orientation. A consistent feature is formation of secondary joints oriented at high angle to the pre-existing joints, initiating during the first 2.4 % displacement (% of layer thickness), and
increasing in number during the experiment. Another consistent feature is formation of conjugate faults. However, they show a wider range of initiation time, starting at 3.8 % displacement (12° JF-angle) up to 11.8 % displacement (16° JF-angle). We note that onset of the formation of conjugates is not related to the JF-angle but varies randomly (see image series of experiments in supplement). A third consistent observation is that
fault localization starts in the footwall and propagates stepwise towards the hanging-wall, always localizing at and reactivating pre-existing joints (Fig. 6).

All experiments share a curvature of the fault scarp towards the footwall at the boundaries, which is a boundary effect caused by the design of the deformation box, similar to what has been observed in the experiment without pre-existing joints (see above).

## SED

doi:10.5194/se-2015-131

### Dilatant normal faulting in jointed cohesive rocks

M. Kettermann et al.

Friction on the sidewalls of the box between the pre-defined 60° fault and the fault localizing at the 90° dipping vertical joints causes material to break off (red arrows in Fig. 5). This effect is limited to the outermost few centimeters of the model and is therefore interpreted as an artifact caused by the boundary condition and is not included in the interpretation.

A variable feature of increasing importance with JF-angle is localization of faults at pre-existing joints, i.e. reactivation of joints. In the experiment with 0° JF-angle the fault never cuts through the material between joints but only jumps from joints in the footwall towards joints in the hanging wall (Fig. 5a). With increasing JF-angle the master faults as well as the conjugates form step-overs between individual joints with fracture orientations at a high angle to the pre-existing joints (e.g. Fig. 5d). The fault reactivates pre-existing joints and shows therefore a distinct deviation from the basement fault strike. At higher JF-angles, the fault connects increasingly more pre-existing joints via step-overs (Fig. 7). The main structural and geometrical features observed at the master fault such as step-overs and distribution of strain over different fault strands and reactivated joints occur in the same way at the conjugates, although with less displacement and therefore less prominent.

At step-overs the fault does not localize at the base of the joints but forms a wedge shaped structure (Fig. 8a). This is because the fault cannot change its position abruptly but forms a hard-link ((Peacock and Sanderson, 1991). Additionally, where the fault cuts through un-fractured material, rubble forms and falls into the opening voids.

An additional feature that occurs in experiments with high JF-angle after about 11.8 % of displacement is reverse faulting within the graben, striking roughly orthogonal to the basement fault strike. As the reverse faults form from bottom to top and do not necessarily propagate to surface, the related surface expression is difficult to see in photographs. However PIV analyses displaying the y-component of the displacement field clearly show locations of compression that lead to reverse faulting (Fig. 8b). Since at the pre-cut bounding walls the 60° basement fault angle enforces a sharp discontinuity to the powder, close to the bounding walls antithetic faults accommodate less

SED

doi:10.5194/se-2015-131

**Dilatant normal faulting in jointed cohesive rocks**

M. Kettermann et al.

**SED**

doi:10.5194/se-2015-131

**Dilatant normal faulting in jointed cohesive rocks**

M. Kettermann et al.

displacement and subsidence in the graben is small. In the center of the box, however, the fault develops freely and dips steep in the upper parts of the powder column, producing more offset on the antithetic faults and hence a higher subsidence within the graben. This subsidence-gradient produces a space-problem which results in formation of reverse faults. However, we observed reverse faults with minor displacements in only two experiments (20 and 25°) and they are accompanied by extensional fractures, which allow us to assume no important effect of the reverse faults on the studied features.

## 4  Quantitative analysis of the analogue models

In order to quantify the effect of JF angle, we carried out analysis of the following measureable parameters using interpreted map view images (Fig. 9): Maximum damage zone width, area fraction of open gaps, degree of segmentation, number of secondary fractures and number of connected pre-existing joints within the damage zone. For quantifying damage zone width, we measure the maximum distance between the unfractured parts of the host rock around the master fault. To measure the area fraction of open gaps, we traced the open fracture networks and quantified their percentage of bulk area using the ImageJ software (Abràmoff et al., 2004). Degree of segmentation is the total number of pre-existing joints accommodating strain. Eventually, we measure the angles between pre-existing joints and secondary fractures using ArcMap software (ESRI – Environmental Systems Resource Institute, 2014).

Our quantitative analyses show an increase of all analyzed attributes from small to large JF-angles for angles higher than 8° (Fig. 10). That no change in structural style between JF-angle of 0 and 4° is observed is possibly an effect of the limited width of the deformation box, as in experiments with small joint-fault angles joints do not necessarily intersect the basement fault trace. In addition to these general trends we note that the area fraction of open fractures increases by only 3 % and varies throughout the experimental series. The increasing trend is most pronounced in the number of

SED

doi:10.5194/se-2015-131

**Dilatant normal faulting in jointed cohesive rocks**

M. Kettermann et al.

secondary fractures, the number of connected joints and the degree of segmentation, which increases by over 150, about 100 and about 130 %, respectively. Interestingly, the secondary fractures are more abundant in the footwall. However, in the experiment without pre-existing joints we count more than 40 secondary fractures and a damage zone width of 13.5 cm, both exceeding all measured values of experiment with pre-existing joints, while the area fraction of open gaps with 5.2 % is smaller (data points are marked with filled square, circle and star in Fig. 10).

Rose diagrams plotting pre-existing joints and secondary fractures show that orientation of secondary fractures is always at a high angle to joint strike (Fig. 11). Only in the experiment with JF-angle of 8° this relationship is not obvious. Overall, we observe that the main fault gap is increasingly filled with rubble with increasing JF-angle.

## 5   Discussion – faulting in jointed rocks

### 5.1   Deformation at different angles

Our experiments provide insights on how pre-existing joints influence normal faults in nature. In our experiments, the most counterintuitive result is the observation that most of the secondary fractures initially occur in the footwall of the normal fault, rather than in the hanging wall, where most strain is accommodated at a later stage. This implies that deformation initiates in the footwall, probably at relatively long distance with respect to the normal fault (few cm). During ongoing deformation, the secondary fractures gradually step over into the hanging wall, until a steady state with mostly hanging wall deformation is reached. Figure 12 shows six PIV images of the experiment with 12° JF-angle illustrating the progressive evolution of a fault at 2, 9, 13, 23, 42 and 14.7 % displacement. Therefore, if a fault system is still evolving, major fluid pathways are located in the foot wall, whereas in long-lived steady state fault systems substantial additional fluid pathways are created in the hanging-wall of the master fault.

The second important observation is that connectivity of the joints increases with increasing JF-angle. This rather straight-forward result has likewise large implications on fluid flow through the system, as connectivity and fracture surface increases. Whereas at low JF-angles fluid flow is concentrated to a small area with low connectivity, systems with higher JF-angles provide a wide zone of interconnected fractures. Our study for the first time is able to quantitatively show this connectivity increase and related parameters (Fig. 10). In areas of variable angle between joints and faults, which probably is rather the rule than the exception, this should be considered. Examples for such settings may be the Canyonlands National Park, or carbonate fields of the Middle East (Daniel, 1954).

## 5.2 Comparison to other models

Whereas studies on interaction between dilatant joints and faults are limited, the interaction of multiple stages of shear faulting has been studied in analogue models by several authors. Henza et al. (2010, 2011) performed experiments in which two phases of faulting at different angles were applied. The major difference to our models is the different material: Henza et al. (2010) use wet clay that does not lose cohesion at fractures or faults, whereas we use dry powder forming cohesionless joints and open fractures. The different approaches are valid for different natural examples. In these experiments second phase faulting localizes at first phase faults but also forms new faults. Similarly, map view of the experiments of Henza et al. (2010) and of this study are comparable. The number of newly formed fault segments increases with increasing angle between maximum principal stresses of first and second phase faulting. Our experiments corroborate these findings, as we observe a systematic increase of the number of new formed fractures and fault segments at step-overs. The result is zigzagged map-view fault geometry comparable to this study. However, in the clay experiments by Henza et al. (2010), step-overs do not develop at the high angles we observe. Kattenhorn et al. (2000) showed that the angle of secondary joints is related to the ratio between fault-parallel and fault-perpendicular stress. This stress ratio differs for cohesive faults

Discussion Paper | Discussion Paper | Discussion Paper | Discussion Paper |

**SED**

doi:10.5194/se-2015-131

**Dilatant normal faulting in jointed cohesive rocks**

M. Kettermann et al.

as in the experiments of Henza et al. (2010) and cohesionless joints as in the presented models, explaining the different orientations of secondary fractures.

## 5.3  Comparison to natural examples

Our results have direct implications for our understanding of natural dilatant fault sys-
tems in jointed rocks. In particular, we can make inferences from our models to the Grabens area of the Canyonlands National Park, Utah, USA, which is an archetype for dilatant faults in jointed rocks (e.g. McGill and Stromquist, 1979; Moore and Schultz, 1999; Rotevatn et al., 2009). Here, the northern part of the Grabens is characterized by prominent joint sets, which are older than the formation of the dilatant faults (McGill and Stromquist, 1979; Schultz-Ela and Walsh, 2002). The most prominent joint set con-
sists of several 100s of m long joints and roughly follows a NNE-SSW striking arcuate geometry of the graben bounding faults. Angles between this joint set and fault strike range between 0 and $\sim 25°$ (Kettermann et al., 2015), which is the range covered in our experiments.

In Canyonlands National Park a second set of pre-existing joints exists which is ori-
ented orthogonal to the NNE-SSW striking joint set. This is parallel to the orientation of the developing secondary fractures observed in our analogue experiments and cross-cuts the first set. These joints may be reactivated by normal faulting during slip on the basement faults. This corroborates findings by Cartwright and Mansfield (1998), who report progressive opening of these joints related to normal faulting.

The grabens of Canyonlands National Park developed as an extensional fault array on top of a deforming layer of evaporites. Faults dip at 60–80° below the jointed layer, comparable to our model setup (Kettermann et al., 2015; McGill and Stromquist, 1979; Moore and Schultz, 1999). The following structural elements observed in the exper-
iments are also present and common in the field. The graben walls are surfaces of pre-existing joints at which the faults localize (Kettermann et al., 2015). Comparable to the models, in the field we infer a progressive migration of the graben bounding faults towards the foot wall by reactivating several pre-existing joints before a steady master

Discussion Paper | Discussion Paper | Discussion Paper | Discussion Paper | Discussion Paper |

**SED**

doi:10.5194/se-2015-131

**Dilatant normal faulting in jointed cohesive rocks**

M. Kettermann et al.

fault forms. Where joints are at an angle with respect to the orientation of the grabens, i.e. not normal to the regional direction of extension, faults step over from one joint to another forming the typical zigzagged shape (Fig. 7d).

As graben walls are vertical and faults dip shallower at depth, open fissures form
at reactivated joints. In the field these are mostly filled with rubble and Quaternary sediments but at numerous locations sinkholes resulting from dilatational faulting exist where sediment and rainwater are transported into the subsurface (Kettermann et al., 2015). Ground penetrating radar studies (Kettermann et al., 2015) suggest that the hanging-walls of the graben-bounding faults (i.e. the graben floors) are faulted as well,
which is in agreement with the observations of our models. This shows that our models are capable of correctly reproducing the characteristic features observed in similar natural settings, allowing us in turn to make predictions of natural fault systems from these models. For example, our models suggest, that along the graben-bounding faults in the subsurface interconnected fluid pathways exist, that are partially filled with unce-
mented, coarse grain sediments and rubble. Visual proofs are the sinkholes that occur at several places along faults (Biggar and Adams, 1987; Kettermann et al., 2015).

Another example of normal faulting in pre-fractured cohesive rocks is the caldera collapse in Campi Flegrei, southern Italy. During collapse, faults reactivate steep pre-existing joints, and detailed analysis of the fracture pattern and younger faults shows
that collapse is controlled by the inherited structures (Vitale and Isaia, 2014). This interaction localizes later volcanic activity in areas adjacent to the caldera. Our modeling efforts corroborate these findings, and show that it is formation of step-overs and distribution of strain across several normal faults which cause new craters to form preferentially in areas of high JF-angles.

The rift zone in Iceland shows similar features. Faults often localize along vertical cooling joints resulting in a planar fault geometry with abrupt changes of fault dip rather than a pure listric shape (Angelier et al., 1997). This characteristic fault shape could be observed in the grabens of Canyonlands NP or in faulted basalts on Hawaii (Holland et al., 2006) and in the presented experiments and is more or less independent of the

**SED**

doi:10.5194/se-2015-131

**Dilatant normal faulting in jointed cohesive rocks**

M. Kettermann et al.

angle between joints and faults. Holland et al. (2006) and Holland et al. (2011) propose a connectivity of open fractures along faults up to great depths based on field and laboratory observations. Our models suggest that this connectivity can be enhanced by the existence of pre-existing vertical joints.

However, the presented results are valid only for pure dip-slip normal faulting. Oblique faulting can produce similar structures without pre-existing joints as shown by Grant and Kattenhorn (2004) in the rift zone on Iceland. Here, vertical joints in an angle with respect to the general fault strike trend are formed in the very early stages of deformation. The resulting structures are mostly comparable to the ones described in this paper, but the temporal and genetic relationship between faults and joints is different and joints are relatively short in extend as they are related to the local faulting rather than a regional process.

## 6   Conclusions

We studied the influence of pre-existing vertical, cohesionless joints on the development of faults with different angles between both. Robust structural features that occur in the models as well as in field prototypes and similar experiments validate our models. In detail we could show that:

– The damage zone width increases by about 50 % and the secondary fractures within this zone by more than 100 % with increasing JF-angle.

– The map-view area fraction of open gaps increases only 3 % from 0 to 25° JF-angle.

– Antithetic faults show similar geometries and damage zone dimensions as the master fault.

– Secondary joints and step-overs are oriented orthogonal to the primary joint orientation.

**SED**

doi:10.5194/se-2015-131

**Dilatant normal faulting in jointed cohesive rocks**

M. Kettermann et al.

Discussion Paper | Discussion Paper | Discussion Paper | Discussion Paper |

**SED**

doi:10.5194/se-2015-131

**Dilatant normal faulting in jointed cohesive rocks**

M. Kettermann et al.

– Experiments without pre-existing joints show a wider fracture network with a higher fracture density, while at the same time providing less open space. However, due to the length of the pre-existing open joints, areas far beyond the fractured parts are connected to the system.

In summary, the angle between pre-existing joints and faults has a distinct effect on the network of open fractures mostly in terms of fracture surfaces and connectivity, while the volume of open space does not change dramatically. However, fluid pathways are created over a large area which has a strong influence on fluid flow. Structures in our models compare well with field prototypes such as the grabens of Canyonlands NP, suggesting a predictive capability of these models. Investigating the influence of parameters such as joint spacing or dimensions will be part of future work in combination with discrete element models that allow investigating detailed fracture connectivity at depth.

*Acknowledgements.* We like to thank Marc Miller and Vicky Webster from the US National Park Service for their kind support in the preparation of the field study in Canyonlands National Park.

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

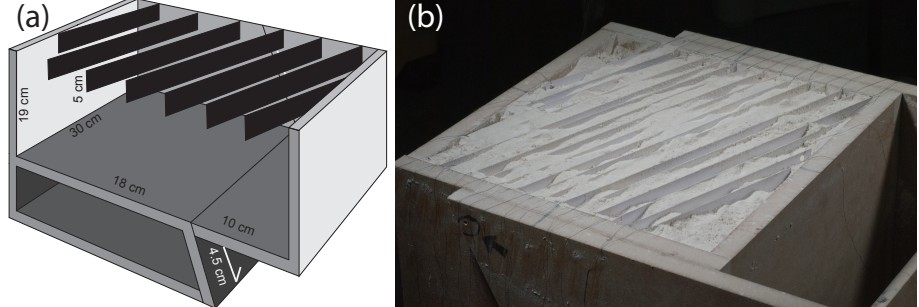

**Figure 1. (a)** Dimension and principle setup of the deformation apparatus. Black bands symbolize paper sheets that are used for joint creation. **(b)** Experiment after sieving in the hemihydrate powder, with the paper sheets still in place. Paper sheets are removed before deformation begins.

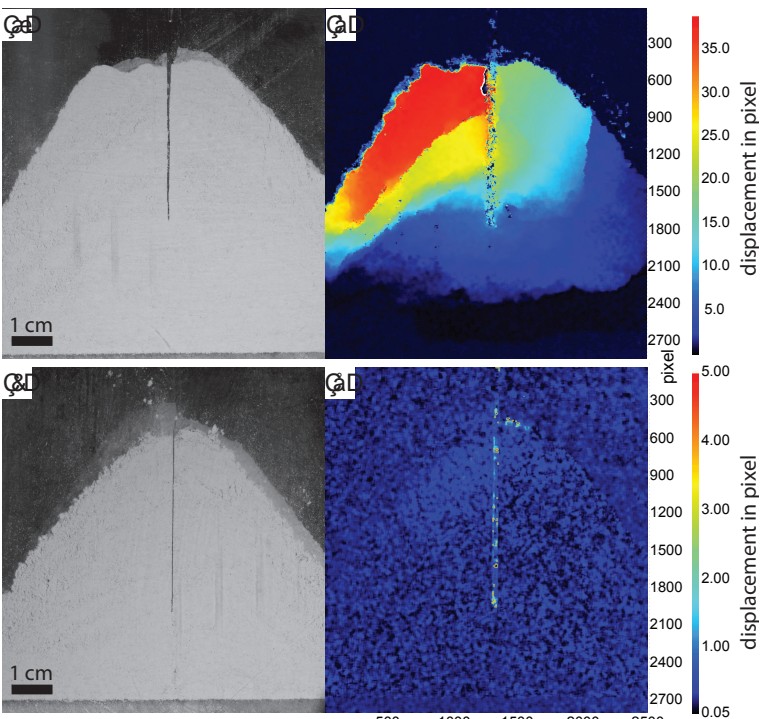

**Figure 2. (a, b)** Raw photo and deformation analysis of a joint in a hemihydrate powder pile created by impressing a blade. The powder is strongly affected. **(c, d)** Raw photo and deformation analysis of a joint in a hemihydrate powder pile created by sieving the powder around a sheet of paper and removing it afterwards (note the different scale bar for displacement). The removing-paper method proves to be the better choice.

Discussion Paper | Discussion Paper | Discussion Paper | Discussion Paper |

**SED**

doi:10.5194/se-2015-131

**Dilatant normal faulting in jointed cohesive rocks**

M. Kettermann et al.

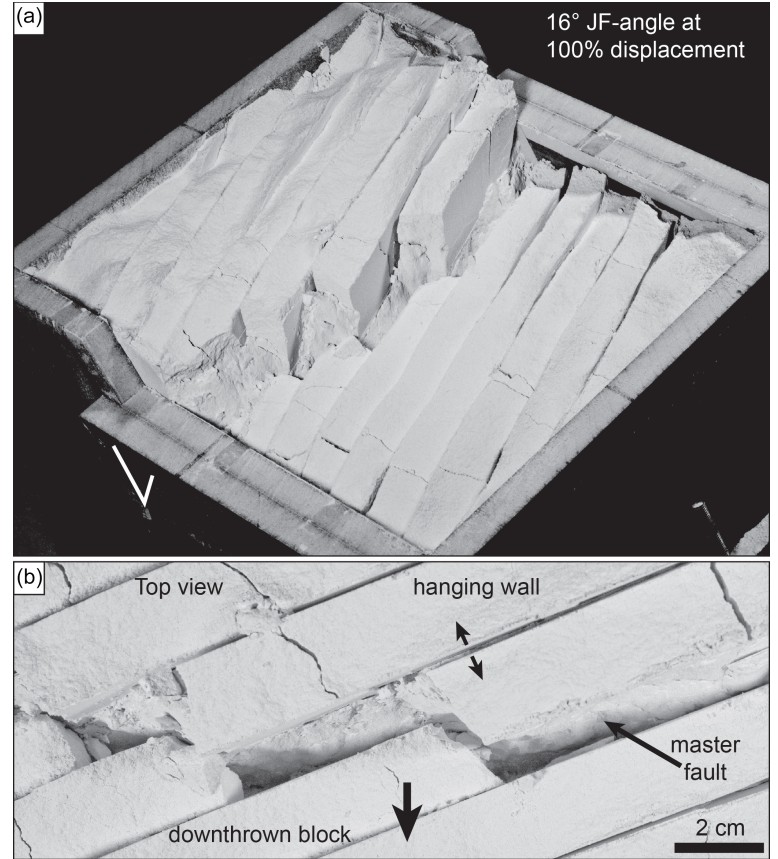

**Figure 3. (a)** Oblique view of the 16° JF-angle showing deformation localized at pre-existing joints and step-over structures. **(b)** Top-view photograph of the same experiment shows the typical zig-zag shape formed by step-overs at the master fault.

Discussion Paper | Discussion Paper | Discussion Paper | Discussion Paper |

**SED**

doi:10.5194/se-2015-131

**Dilatant normal faulting in jointed cohesive rocks**

M. Kettermann et al.

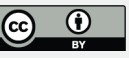

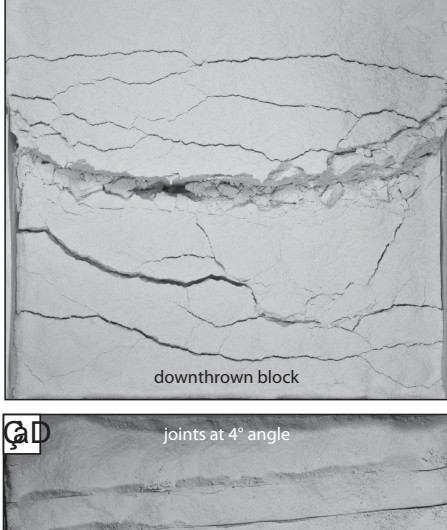

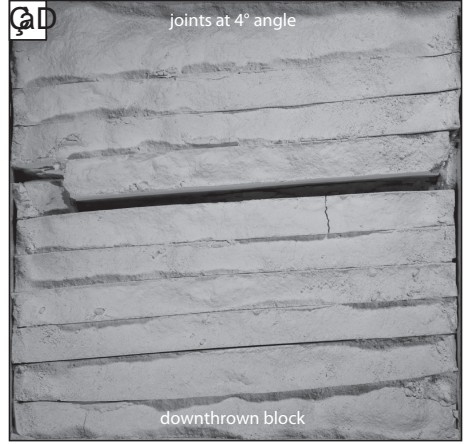

**Figure 4. (a)** Top view photo of an experiment without pre-existing joints. Note that rather rugged shape of the mater fault and the minor fractures. **(b)** Top view photograph of the experiment with a 4° JF-angle. All deformation localizes at the pre-existing joints.

Discussion Paper | Discussion Paper | Discussion Paper | Discussion Paper |

**SED**

doi:10.5194/se-2015-131

**Dilatant normal faulting in jointed cohesive rocks**

M. Kettermann et al.

## SED

doi:10.5194/se-2015-131

**Dilatant normal faulting in jointed cohesive rocks**

M. Kettermann et al.

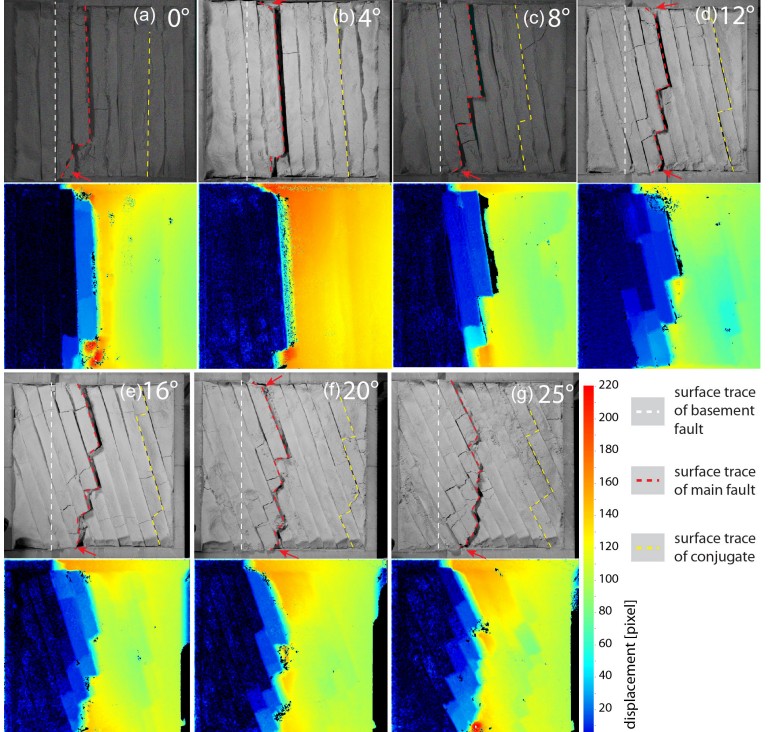

**Figure 5.** Map-view photographs of the experiment series at maximum displacement. Red lines mark the master fault; yellow lines mark the main antithetic fault. White lines illustrate the extent of the basement fault at the surface. For each experiment we show a respective PIV image illustrating the total deformation in map-view. Color code gives the displacement in pixels. Note that different blocks experienced different amounts of displacement, while localization is always at pre-existing joints.

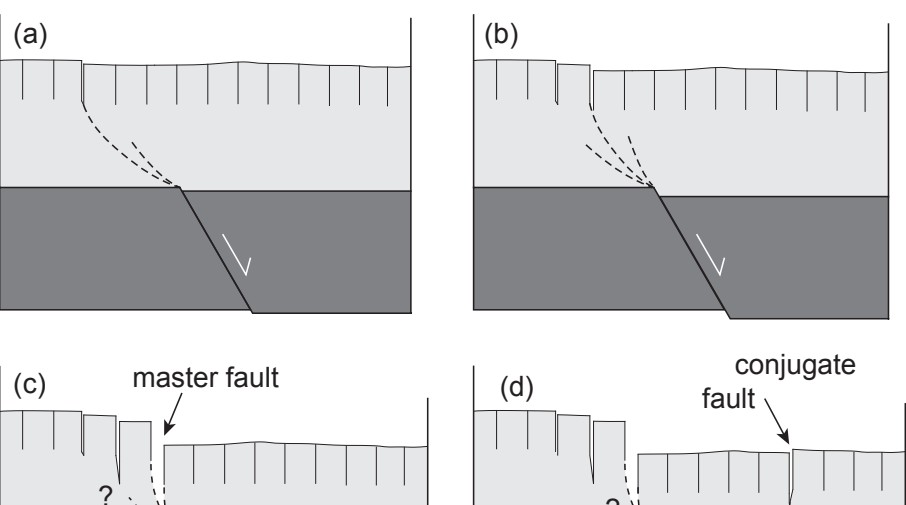

**Figure 6.** Conceptual sketch illustrating the development of a typical joint controlled fault zone in side-view.



**SED**

doi:10.5194/se-2015-131

**Dilatant normal faulting in jointed cohesive rocks**

M. Kettermann et al.

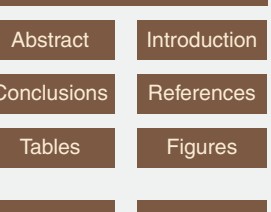

Title Page

| Abstract | Introduction |
| Conclusions | References |
| Tables | Figures |

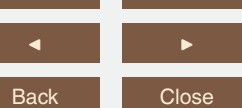

Back | Close

Full Screen / Esc

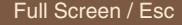

Interactive Discussion

**SED**

doi:10.5194/se-2015-131

**Dilatant normal
faulting in jointed
cohesive rocks**

M. Kettermann et al.

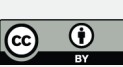

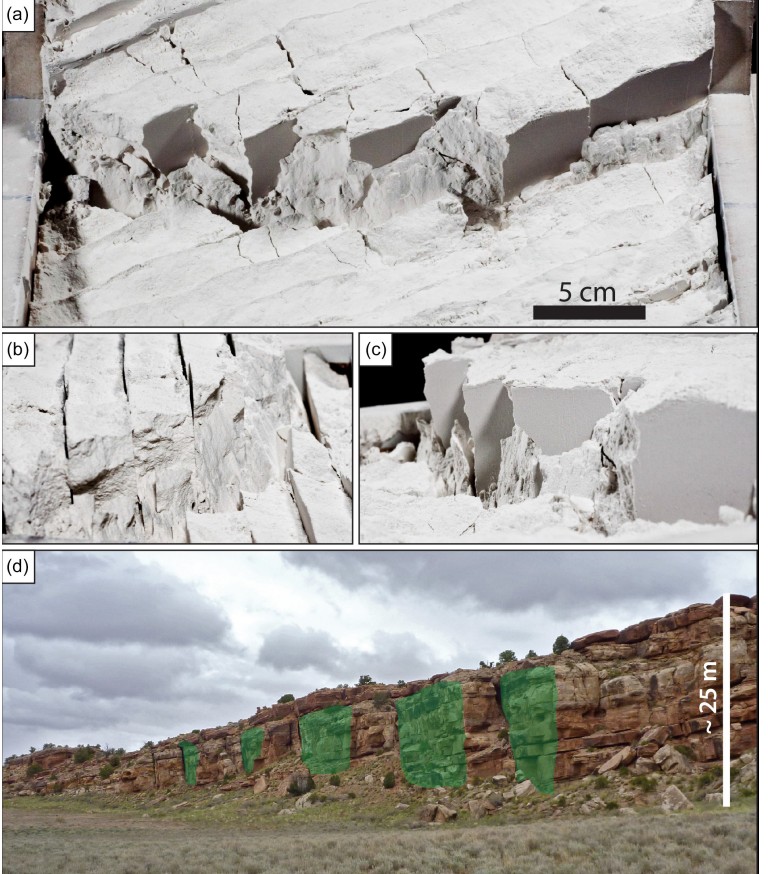

**Figure 7. (a)** front view of the experiment with 25° JF-angle. **(b)** View from left side. **(c)** View from right side. **(d)** Comparable structures in Canyonlands NP. Green areas mark joint surfaces.

**SED**

doi:10.5194/se-2015-131

**Dilatant normal faulting in jointed cohesive rocks**

M. Kettermann et al.

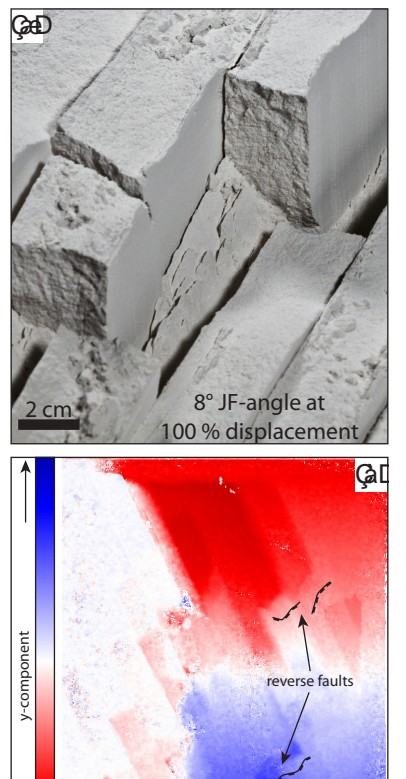

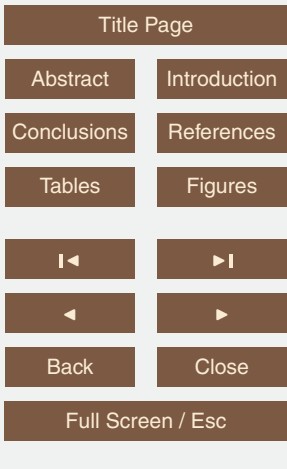

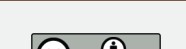

**Figure 8. (a)** Wedge shape at a fault step-over; **(b)** Reverse faults in the hanging wall can be best shown by PIV images. Color code gives the *y*-component of the deformation. Sharp changes in color intensity indicate compression or dilation.

**SED**

doi:10.5194/se-2015-131

**Dilatant normal faulting in jointed cohesive rocks**

M. Kettermann et al.

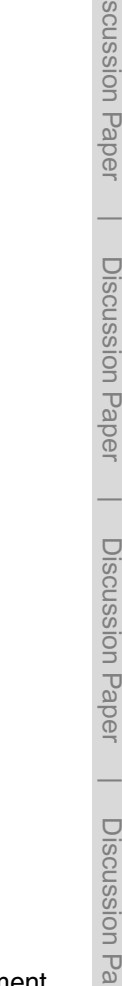

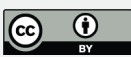

fractures opened during faulting

5 cm

**Figure 9.** Top-view image of interpreted newly opened fractures at maximum displacement, exemplary of the 16° JF-angle experiment.

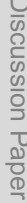

**SED**

doi:10.5194/se-2015-131

**Dilatant normal faulting in jointed cohesive rocks**

M. Kettermann et al.

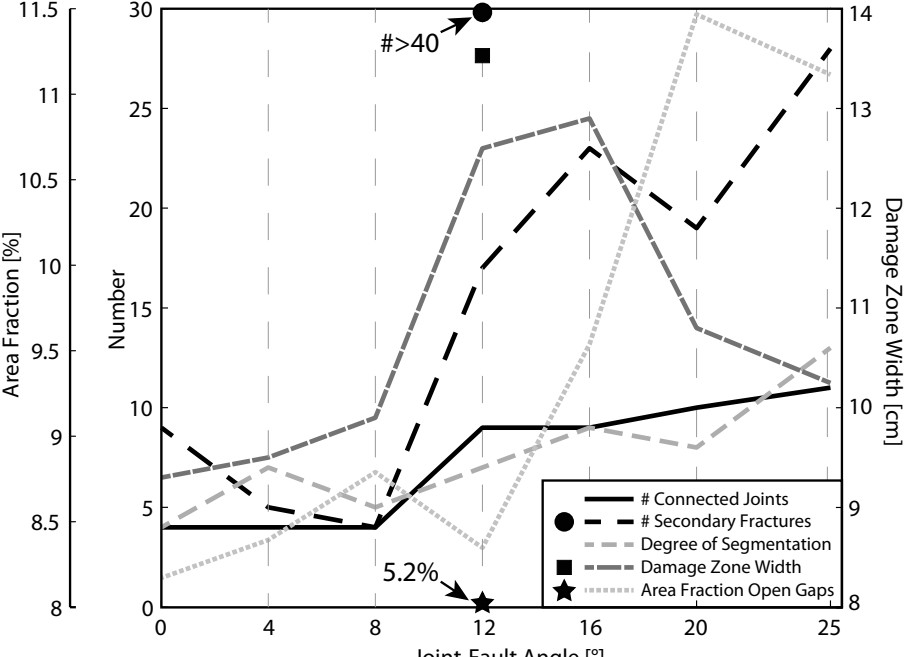

**Figure 10.** Results of the quantitative analysis. For definitions of the individual parameters please refer to Sect. 3.1.

Discussion Paper | Discussion Paper | Discussion Paper | Discussion Paper

**SED**

doi:10.5194/se-2015-131

**Dilatant normal faulting in jointed cohesive rocks**

M. Kettermann et al.

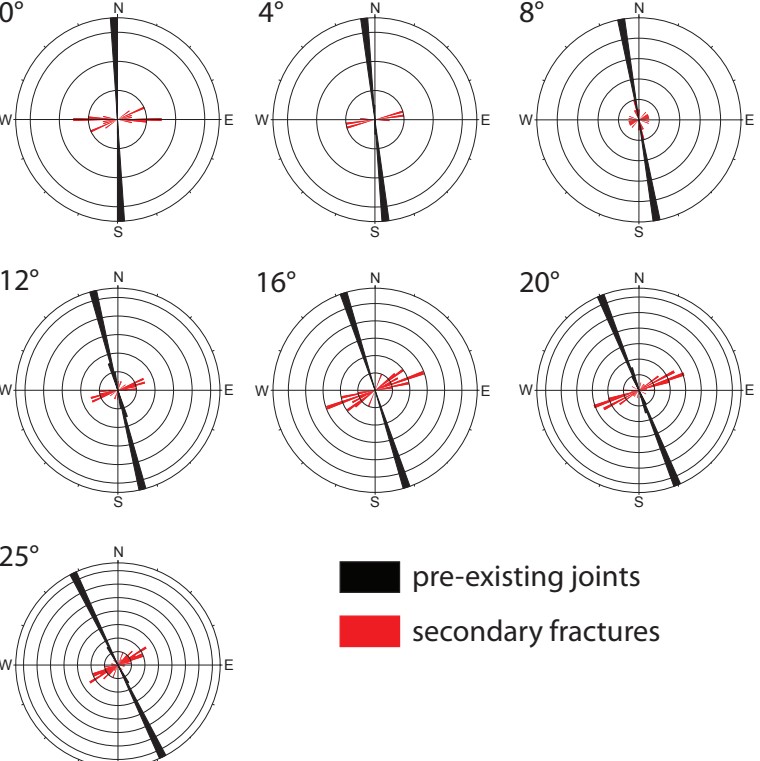

**Figure 11.** Roseplots showing the orientation of pre-existing joints (black) and secondary fractures (red) for all experiments. Strike direction of the basement fault is N-S. Note that secondary fractures are always in a high angle to the pre-exiting joints.

Discussion Paper | Discussion Paper | Discussion Paper | Discussion Paper |

**SED**

doi:10.5194/se-2015-131

**Dilatant normal faulting in jointed cohesive rocks**

M. Kettermann et al.

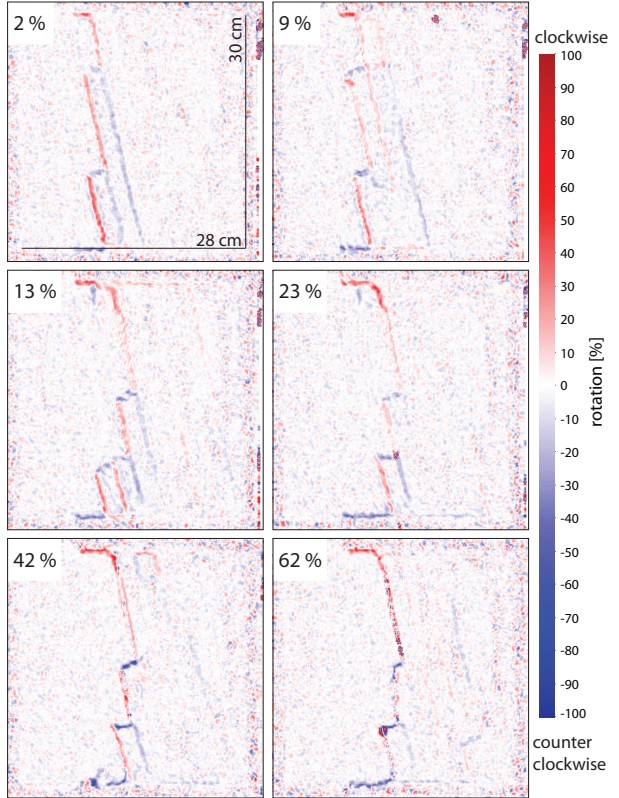

**Figure 12.** PIV images series of the 12° JF-angle experiment showing how different joints are reactivated at different times during deformation.