# Peer review of "Dilatant normal faulting in jointed cohesive rocks: a physical model study"

_Solid Earth, 2015_

## Referee Comment (RC1) · A. Billi (Referee) · 21 Jan 2016

Dear Editor,

I have now revised with much interest the manuscript by Kettermann et alii on dilatant normal faulting in jointed cohesive rocks. I have found this manuscript in a very nice shape. It is well written, clear, and scientifically sound. It addresses (using physical modeling) a classical but still poorly understood problem in brittle structural geology, that is the relationship between normal faulting and pre-existing joints.

Some very interesting observations and conclusions are drawn, such as enhanced jointing in the fault footwall during the early stages of normal faulting (and then migration toward the hangingwall) or enhanced fracture connectivity with larger angles between pre-existing joints and the normal fault. Comparisons with natural examples

are very appropriate and useful for a better understanding of the problem and results.

One problem I have had is that I was unable to access the data repository to check the supplemental material.

I have very few comments and think that the paper could be published as it is or almost.

I see that the core/synthesis of all the work done is in the diagram of Fig. 10, where a series of features/parameters such as joint connectivity, degree of fault segmentation, etc are graphed against the angle between pre-existing joints and the main (developing) normal fault. From the text (sections 3 and 4) and the figures (Figs 2-5) I cannot well understand how most of these parameters are measured (e.g. joint connectivity) and where the raw data are stored/shown. Is there a synoptic table with all the experiments and related data? Further information about measurements and data will be very important to complete this work, at least in my opinion. I refer to enhanced figures, a synoptic table of data, and more text to better explain data acquisition.

Eventually, I would like to acknowledge that I am not expert of physical/analog experiments, related materials, rheologies, etc. Maybe a reviewer expert in this field could provide important comments on this side of the work.

Sincerely Andrea Billi
* * *

---

## Referee Comment (RC2) · O. Galland (Referee) · 22 Feb 2016

Review of Kettermann et al manuscript

This manuscript deals with the laboratory study of the structural features associated with rifting of jointed rock formations. The first section nicely introduces the problem, highlighting that most former research focused on rifting of intact rocks, whereas it is known that natural rocks are intensely jointed/fractured. Therefore this study is innovative and relevant.
The method is also well described, and clear. It explains in a very didactic manner how the experiments are prepared, describing in details the technical tests the authors performed. In particular, they discuss the methods used to prepare pre-joints in their experiments, and justify the method they chose.
The results are also convincing. They show how the angle between extension direction and the orientation of pre-existing joints control the deformation patterns. They also highlight very well that the deformation pattern strongly differs, whether the models are pre-jointed or intact. This fact demonstrates that former intact models are not relevant for studying the natural complexity of rifted rock formations.

I notice, however, some critical potential problems in the concepts of the presented experiments. Indeed, it is not clear how reproducible the experimental results are, especially with respect to the initial conditions (see details below). In addition, the authors arbitrarily selected a joint depth, without justifying it and discussing its relevance/effect.

Therefore, I would recommend the manuscript to be accepted for publications, after some critical corrections.

**Main comments**

*Reproducibility of the results with respect to initial conditions.* Figure 10 shows complex trends of the results. In particular, the authors argue that the lack of trend for the JF-Angle between 0 and 8 is possibly an "effect of the limited width of the deformation box, as in experiments with small joint-fault angles joints do not necessarily intersect the basement fault trace". This is critical, because the authors implicitly imply that according to the initial conditions, i.e. the position of the joints with respect to the basement fault trace, the results are not reproducible. This means that the initial positions of the joints with respect to the basement fault trace is a very critical factor. And I suspect that it might be also the case for JF-Angle larger than 8 degrees. Indeed, depending on the positions of the joints, a different number of joints can intersect the basement fault trace. This fact strongly questions the relevance of the "trends" of Figure 10. At least, this critical initial condition likely explains the very scattered and chaotic results displayed in Figure 10. The authors should really discuss, and even quantify, the effect of the initial position of the joints with respect to the basement fault trace. I suspect that this would require other laboratory tests to be performed.

*Initial conditions*. The heights of the joints are of 5 cm, with respect to the 19 cm thick box. What is the effect of the ratio between the heights of the joints and the total thickness of the box? What would happen if, for example, the joints are as deep as half of the box? This has major consequences, as joints often fully cross cut rock layers entirely. I suspect that the results would be very different. The authors should also discuss this very important initial condition and it potential effect on their results. I recommend the authors to perform a few experiments with varying heights of the pre-existing joints.

*Section 5.3, Field data*. This section is difficult to read, as there is no graphic support, i.e. no figure. The only figure is a field photograph, where none of the conclusions of the laboratory study is visible. This is incompatible with the conclusion of the authors that states "Robust structural features that occur in the models as well as in field prototypes…". In this paper, it is impossible to assess the relevance of the laboratory results with respect to field data. I thus would recommend the authors to include a figure (at least) displaying field data, such that the comparison between laboratory and field results is more obvious.

**Minor comments**
Page 3, line 6. The authors could also include references to Galland et al. (2006), Galland et al. (2007) and Le Corvec et al. (2013) at the end of the last sentence of the paragraph.

Page 5, lines 7-18. The authors should specify whether the optical distortion is corrected or not. This has important implications on angle measurements presented later. Also they should indicate the lens characteristics, especially focal length, such that the reader has a good idea of the amount of optical distortion to be corrected.

Section 3. In general, it lacks lots of references to figures. It is difficult to follow the text. Therefore, almost for each statement of the text, the authors should refer to the corresponding figure.

Page 7, last sentence. I don't understand the sentence. Maybe I am not awake, but the structure seems quite complicated. It would be good to split the sentence into several.

Page 7, lines 26-27. The authors mention that the y-displacement components highlight reverse faults. This is the case because the y-axis is coincidently properly oriented with respect to the local shortening. In addition, without the drawings of the authors, the reader would likely not see the reverse faults. In general, it is better to compute the divergence of the displacement field: positive divergence means extension, negative divergence means compression (Byrne et al. 2015). I recommend the authors to plot the divergence field to highlight the local compression.

Figure 12. What is actually plotted in this figure? Shear strain, divergence, rotation angle? This must be specified.

References

Byrne, P.K., Holohan, E.P., Kervyn, M., van Wyk de Vries, B. & Troll, V.R. 2015. Analogue modelling of volcano flank terrace formation on Mars. *Geological Society, London, Special Publications*, **401**, 185-202, doi: 10.1144/sp401.14.

Galland, O., Cobbold, P.R., de Bremond d'Ars, J. & Hallot, E. 2007. Rise and emplacement of magma during horizontal shortening of the brittle crust: Insights from experimental modeling. *Journal of Geophysical Research*, **112**, doi: 10.1029/2006JB004604.

Galland, O., Cobbold, P.R., Hallot, E., de Bremond d'Ars, J. & Delavaud, G. 2006. Use of vegetable oil and silica powder for scale modelling of magmatic intrusion in a deforming brittle crust. *Earth and Planetary Science Letters*, **243**, 786-804.

Le Corvec, N., Menand, T. & Lindsay, J. 2013. Interaction of ascending magma with pre-existing crustal fractures in monogenetic basaltic volcanism: an experimental approach. *Journal of Geophysical Research: Solid Earth*, **118**, 968-984, doi: 10.1002/jgrb.50142.

---

## Author Comment (AC1) · 22 Mar 2016

**Answer to the comments of Andrea Billi**

Dear Andrea Billi,

Thank you very much for your kind review of our manuscript and the helpful comments. In the following we answer to each issue individually:

1. *One problem I have had is that I was unable to access the data repository to check the supplemental material.*

→ We agree with the reviewer that access to the supplementary material is not perfectly solved. This is mostly due to the large amount of video files. However we finally found a way to publish the videos as one dataset with one DOI using Pangea Data Publisher for Earth & environmental Science. We added the following sentence to the text and updated further references:
**"Movies produced from image series of all experiments and the respective PIV images are freely accessible at https://issues.pangaea.de/browse/PDI-11894"**

The data publishing is however still in progress. Until the videos are finally published with open access with an assigned DOI (which will be the case for the final version of the manuscript) the videos can be found here:
https://rwth-aachen.sciebo.de/index.php/s/adccOLuVPT2dk63

2. *I see that the core/synthesis of all the work done is in the diagram of Fig. 10, where a series of features/parameters such as joint connectivity, degree of fault segmentation, etc are graphed against the angle between pre-existing joints and the main (developing) normal fault. From the text (sections 3 and 4) and the figures (Figs 2-5) I cannot well understand how most of these parameters are measured (e.g. joint connectivity) and where the raw data are stored/shown.*

→ This is a valid comment. In the new version of the manuscript we include pictures and interpretation in the appendix. We added a sentence to the manuscript: "**Pictures of the experiments and their interpretation can be found in the appendix.**"

3. *Further information about measurements and data will be very important to complete this work, at least in my opinion. I refer to enhanced figures, a synoptic table of data, and more text to better explain data acquisition.*

→ In the new version of the manuscript we also add a table summarizing our data. For showing better data acquisition, we added to figure 9 how we performed interpretation following basic routines. Additionally we added some details on methods in the text ("Section "Quantitative analysis of the analogue models" to clarify the measurement workflow. It now reads:

**"In order to quantify the effect of JF angle, we carried out analysis of the following measureable parameters using interpreted map view images (see Fig. 10 for interpreted map and illustration of measured parameters): Maximum damage zone width, area fraction of open gaps, degree of segmentation, number of secondary fractures and number of connected pre-existing joints within the damage zone. For quantifying damage zone width, we measure the maximum distance**

between the un-fractured parts of the host rock around the master fault (see Fig. 10). In cases where damage by the main fault cannot be separated from damage by the antithetic fault, half the distance between both is assumed as damage zone boundary. To measure the area fraction of open gaps, we manually traced the open fracture networks and quantified their percentage of bulk area using the ImageJ software (Abràmoff et al., 2004). Degree of segmentation is the total number of pre-existing joints accommodating strain, which was determined using PIV analysis. Eventually, we measure the angles between pre-existing joints and secondary fractures using ArcMap software (ESRI - Environmental Systems Resource Institute, 2014). Top-view photographs of all experiments and their interpretation can be found in the appendix. Table 2 summarizes the measured data."

---

## Author Comment (AC2) · 22 Mar 2016

Answer to the review of Olivier Galland

Dear Olivier Galland,

Thank you very much for your thorough review and critical comments. Incorporating corrections accordingly to your suggestions will strongly improve the quality of the manuscript. Below we answer to your individual comments in detail and provide changes made.

Kind regards,

Michael Kettermann and colleagues

1. Reproducibility of the results with respect to initial conditions. Figure 10 shows complex trends of the results. In particular, the authors argue that the lack of trend for the JF---Angle between 0 and 8 is possibly an effect of the limited width of the deformation box, as in experiments with small joint---fault angles joints do not necessarily intersect the basement fault trace. This is critical, because the authors implicitly imply that according to the initial conditions, i.e. the position of the joints with respect to the basement fault trace, the results are not reproducible. This means that the initial positions of the joints with respect to the basement fault trace is a very critical factor. And I suspect that it might be also the case for JF-angles larger than 8 degrees. Indeed, depending on the positions of the joints, a different number of joints can intersect the basement fault trace. This fact strongly questions the relevance of the trends of Figure 10. At least, this critical initial condition likely explains the very scattered and chaotic results displayed in Figure 10. The authors should really discuss, and even quantify, the effect of the initial position of the joints with respect to the basement fault trace. I suspect that this would require other laboratory tests to be performed.

→ This is a very good comment. We looked into this and found that in fact the influence of the discussed initial conditions is not as strong as suspected. With a JF-angle of 0° the initial position of the joints only becomes critical for very large joint spacings, while in the presented geometry the fault truncates the joints at every possible position. The 4° JF-angle is in fact critical since the possible number of JF intersects can be 0 or 1 depending on the initial position. A substantially wider box would result in intersections and possibly the formation of stepovers. This cannot be represented in our data. However, at and above JF-angles of 8° the geometry always provides at least two intersections, independent of location of the joints with respect to basement fault. This means that we can always observe joint-fault interaction at two independent points, and we argue that the presented values are thus representative. However, the reviewer is right with stating that data in the range 0°>x°